# Synthesis and Properties of Biodegradable Hydrogel Based on Polysaccharide Wound Dressing

**DOI:** 10.3390/ma16041358

**Published:** 2023-02-06

**Authors:** Meiling Shao, Zhan Shi, Xiangfei Zhang, Bin Zhai, Jiashu Sun

**Affiliations:** College of Chemistry and Chemical Engineering, Shangqiu Normal University, Shangqiu 476000, China

**Keywords:** degradable hydrogel, wound dressing, bacteriostatic performance

## Abstract

The metabolic disorder of the wound microenvironment can lead to a series of serious symptoms, especially chronic wounds, which result in significant pain in patients. At present, there is no effective and widely used wound dressing. Therefore, it is important to develop new multifunctional wound dressings. Hydrogel is an ideal wound dressing for medical nursing because of its abilities to absorb exudate and maintain wound wetting, its excellent biocompatibility, and its ability to provide a moist environment for wound repair. Because of these features, hydrogel overcomes the shortcomings of traditional dressings. Therefore, hydrogel has high medical value and is widely studied. In this study, a biodegradable hydrogel based on polysaccharide was synthesized and used as a wound dressing. The swelling degree and degradability of hydrogel were characterized as the characteristics of the wound dressing. The results showed that the prepared hydrogel was degraded with trypsin and in the soil environment. Furthermore, the wound dressing can effectively inhibit the bacterial environment, promote the deposition of the collagen structure of the wound tissue, and accelerate the healing of the wound. The proposed hydrogel has value in practical medical nursing application.

## 1. Introduction

Hydrogel is a kind of hydrophilic three-dimensional network structure that has water as a dispersion medium, and is insoluble but can swell [1,2]. Under the influence of water, hydrogel can swell and handle large amounts of biological liquid, protecting the network structure within the cross-linked polymer chains and providing a springy effect for the hydrogel-treated objects [3]. A variety of hydrophilic groups are attached to the polymer side chain skeleton in the hydrogel, including –COOH, –OH, –SO_3_H, –NH_2_, and –CONH, which can maintain the water retention property of hydrogel [4]. As a kind of extracellular matrix (ECM) polymer, hydrogel has good biocompatibility, and has been widely used in biomedicine, tissue engineering, and other fields [5,6,7].

At present, wound dressings are mainly divided into traditional wound dressings and new wound dressings. Traditional wound dressings, including hemostatic gauze and bandages, are mainly used to keep the wound dry, absorb the wound exudate, and prevent wound infection [8,9]. However, they easily adhere to the wound and cause secondary damage without an antibacterial effect. In view of the deficiencies of traditional wound dressings, a new wound dressing was designed according to the theory of wet wound healing. Hydrogels have good toughness and can be tightly fitted to the skin as a wound barrier to prevent wound infection [10,11,12]. In addition, studies have shown that the hemostatic property of hydrogels is not only supported by the physical sealing, but also the enrichment of coagulation factors through the absorption of wound extract [13,14]. Hydrogels are used in the preparation of new wound dressings. With the development of preparation technology, biodegradable hydrogels have been developed and applied to wound excipients [15]. Polysaccharide-based hydrogels composed of chitosan and other natural biological polysaccharides (such as sodium alginate and cellulose) have been widely investigated due to their excellent functions [16]. Alginate and chitosan can form a polyelectrolyte complex through the interaction between the carboxyl group of alginate and the amino ions of chitosan. Chen et al. [17] prepared citric acid-modified chitosan hydrogels loaded with tetracycline hydrochloride by freeze–thaw treatment. Antibacterial experiments showed that the hydrogel had an obvious antibacterial effect on both Gram–negative bacteria (Escherichia coli) and Gram–positive bacteria (Staphylococcus aureus). In addition, the tensile strength of the hydrogel was 3 ± 1.5 MPa, the elastic modulus was 2.16 ± 0.1 MPa, and the hydrogel had good tensile properties and toughness. Wound healing experiments showed visible granulation tissue formation 12 days after covering the wound with the hydrogel. Ehteram et al. [18] prepared functional hydrogels loaded with A–tocopherol (vitamin E) by crosslinking chitosan and sodium alginate. The therapeutic effect of the developed hydrogel was studied in a full–layer skin wound model [19]. The experimental results showed that the prepared chitosan/sodium alginate hydrogel dressing had a higher wound healing rate than the wound treated with gauze. Khomsan et al. [20] used the freeze–thaw treatment in the preparation of chitosan/poly (vinyl alcohol)/ZnO hydrogels. The average aperture in the hydrogel was 13.7 ± 5.9 μm, the bibulous rate reached 850%, and the mechanical properties increased with the increase in the elongation of the break. The antibacterial experiment showed that the hydrogel had an excellent antibacterial effect against Gram–positive bacteria (Staphylococcus aureus). In vitro wound healing experiments showed that it had good biocompatibility and could effectively accelerate wound healing. Rezaei et al. [21] synthesized a heat–responsive chitosan (TCTS) hydrogel and loaded it with different concentrations of antimicrobial peptides to create an antibacterial wound dressing against drug–resistant bacteria. The physicochemical properties, release behavior, biocompatibility, and antibacterial activity of AMP–TCS hydrogel against standard strains and drug–resistant Acinetobacter baumannii were determined by in vitro antibacterial experiments. On day 1, AMP–TCTS AMP gel explosive released about 40% of the water, which can occur within 7 days of controlled drug release. The hydrogel was observed to be extremely absorbent for the first 4 h, and absorbency then continued in a steady manner for 10 h. AMP–TCS hydrogel showed good biocompatibility with human fibroblasts. TCTS hydrogels showed no antibacterial activity against standard strains and clinical isolates. By comparison, AMP–TCTS hydrogels showed strong antibacterial activity against standard strains, but only 16 μg/mL hydrogels showed antibacterial activity against drug–resistant Baumanella. The experimental results showed that 16 μg/mL of AMP–TCTS hydrogel could be used as an excellent antibacterial wound dressing against resistant Baumanniella, and this hydrogel is expected to be further tested in clinical trials. Song et al. [22] developed a multifunctional adhesive biohydrogel consisting of 3, 4–dihydroxyphenylpropionic acid modified chitosan (DCS) and p–hydroxybenzaldehyde modified polyethylene glycol (PEGSH). The biohydrogel, which is a kind of hemostatic material having great potential, showed good tensile properties and coagulation, strong adhesion and self–healing properties, and good cytocompatibility and antibacterial properties. The hydrogel eliminates the need for any additional adhesives and antibacterial agents to overcome the shortcomings of traditional hemostatic agents, which have poor stretching ability and poor self–adhesion in humid environments. Kumar et al. [23] prepared chitosan–gellan hydrogel for drug delivery and tissue engineering. Controlled drug release at low pH can be achieved by adding gellan gum. The research of Lei H et al. [13] found that antioxidant hydrogels can remove excessive reactive oxygen species in chronic wounds to reduce oxidative stress, thus improving the wound microenvironment, promoting collagen synthesis and re–epithelialization, and reducing the pH value of the wound to accelerate healing and reduce infection.

Therefore, in this study, a biodegradable polysaccharide–based hydrogel was synthesized and applied as a wound dressing. Using polyurethane as the base material, the colloid surface is controlled by the amide reaction to ensure compatibility. The mechanism of the hydrogel on wound healing was extensively investigated and it was found that the hydrogel can effectively inhibit bacteria and accelerate wound healing.

## 2. Materials and Methods

### 2.1. Materials and Reagents

Hyaluronic acid (HA), sodium periodate, chitosan (CS), sodium alginate (SA), glycol, and dimethyl sulfoxide were purchased from Aladdin Reagent Co., Ltd., Shanghai, China. The cell culture media were purchased from Fuzhou Aoyan Experimental Equipment Co., Ltd., Fuzhou, China. The chemical structures of the purchased SA and CS are shown in Figure 1, and the molecular weight of CS ranged from 100,000 to 150,000.

### 2.2. Methods

SA and CS having the mass ratio of 2/1 were dissolved in autoclaved deionized water to obtain a final concentration of 40 mg/mL. The solution was placed in a petri dish, kept at 60 °C, and stirred for 90 min. After centrifugation, the sediment was dispersed in the liquid, and sodium hydroxide with a mass concentration of 150 mg/mL was added to adjust the solvent to be alkaline. When the liquid turned yellow, the precipitation produced by repeated washing and centrifugation with acetone was treated by vacuum drying and dissolved in distilled water. Three grams of HA was weighed and placed in a reaction vessel, and 300 mL of deionized water was prepared. After the substance was completely dissolved, 1 g of sodium periodate was added and the reactive substance was continuously quenched with ethylene glycol. The SA, CS, HA, sodium periodate, and ethylene glycol mixture solution was introduced into the mold and polymerized to obtain polysaccharide–based degradable hydrogel. The diagram of the polysaccharide hydrogel synthesized by a crosslinking reaction is shown in Figure 2, in which both SA and CS are integrated into the structure of the polysaccharide–based hydrogel. Scanning electron microscopy was used to observe the polysaccharide–based degradable hydrogel and obtain SEM images of the hydrogel structure. The results are shown in Figure 3.

Figure 3 shows that the prepared uniform hydrogel structure is coated on the PE film with the hydrogel solution by the coater, and the medical polyurethane semi–permeable film containing wavy lines is placed on the other side. The coating thickness of the solution was controlled to be 2.5 mm; the coating component was controlled to be cooled naturally and formed, and then cut into rectangular specimens of the same size. The size of the internal holes was centrally distributed in a range of 5–10 μm. After sealing with a sealer, the sample of water coagulation dressing was prepared by irradiating 25 k Gy in a nuclear magnetic resonance spectrometer.

A sample of the hydrogel wound dressing prepared above was weighed and placed in PBS (pH 7.4) buffers, followed by a GX–2030–type high-temperature circulating chamber [24]. After 12 h of treatment at the constant temperatures of 20, 30, 40, 50, 60, and 70 °C, when the wound dressing reached the swelling equilibrium, the excess water on the surface was absorbed and the quality of the hydrogel wound dressing was determined.

## 3. Results and Discussion

### 3.1. Swelling Degree Analysis

The swelling rate of wound dressing was calculated, and the numerical relationship is expressed as:(1)$$R=\frac{D_{s}-D_{f}}{D_{f}}\times 100\%$$
where R is the swelling rate of wound dressing; D_s_ is the swelling equilibrium mass of the wound dressing sample at corresponding temperature; and D_f_ is the original quality of the wound dressing [7]. The corresponding swelling rates of wound dressings under different temperature conditions are shown in Table 1.

The test temperatures were selected in the ranges of 30 °C to 70 °C and 30 °C to 50 °C degrees, with an increment of 5 °C, and 50 °C to 70 °C degrees, with an increment of 10 °C. It can be seen from Table 1 that the prepared biodegradable polysaccharide–based hydrogel wound dressing has strong hydrophilicity. With the increase in ambient temperature, the swelling rate of wound dressing increased, indicating that the prepared hydrogel has strong absorbability and can meet the requirements of normal wound treatment.

### 3.2. Hydrogel Degradability

In order to investigate the biodegradability of polysaccharide–based biodegradable hydrogels, the mass loss of hydrogels under different concentrations of trypsin and in the soil environment was tested. In addition, the morphologies of the degraded hydrogel were observed by scanning electron microscope. Figure 4a shows the mass loss of hydrogel reduced by trypsin at concentrations of 0.1 and 0.2 mg/mL. It can be observed from Figure 4a that the hydrogel at the concentration of 0.2 mg/mL trypsin decreased the hydrolysis rate faster, and the mass loss rate was 68% after 7 days. The results showed that part of the polymer chain segment of the hydrogel was firstly cut off due to the action of enzymes. The damaged part gradually separated from the structure of the hydrogel network and finally dispersed in the solution, resulting in the continuous reduction in the quality of the residual hydrogel. Figure 4b shows that the mass loss rate of the prepared hydrogel reached 61% after 14 days in the soil environment, which indicates that the prepared hydrogel can also degrade in the soil environment.

The changes in hydrogel surface morphology caused by degradation were characterized by SEM, and the results are shown in Figure 5a–c. The biodegradable polysaccharide hydrogel was highly porous, but the degradation of the amide bond on the polypeptide–based vinyl cross–linking agent chain resulted in the destruction of the hydrogel network structure. As shown in Figure 5c, after degradation treatment for 7 days, the internal structure of the hydrogel wound dressing was decomposed into concave and convex structures by microorganisms in the soil. In addition, the surface of the film changed from smooth to rough, and there were holes of different sizes in the film structure. Microorganisms in the soil destroyed the structure of the dressing film, resulting in a compositional degradation process. These results indicate that the prepared biodegradable polysaccharide–based hydrogel is biodegradable and environmentally friendly, which is in line with the application requirements of medical dressings.

### 3.3. Bacteriostatic Performance

A test medium was prepared to test the bacteriostatic performance of the polysaccharide-based degradable hydrogel wound dressing. Nutrient AGAR medium was selected; 5 g of beef paste was prepared, to which 10 g peptone of was added, and the result was stirred evenly. Five grams of sodium chloride was added, followed by an appropriate amount of distilled water to dissolve into a 1000 mL solution. A concentration of 0.1 mol/L sodium peroxide was then added to adjust the pH of the solution to weak acid. The result was placed in a sterilization box and treated for 30 min. When configuring the buffer of the medium, 2.5 g disodium hydrogen phosphate and 1.3 g of potassium dihydrogen phosphate were added to 500 mL of distilled water. After it was fully dissolved, the pH value of the mixed solution was adjusted to about pH 7.4 with hydrochloric acid. After sterilization, the solution was fused with the medium with the above configuration, and Escherichia coli were inoculated on the inclined plane of the medium. The medium containing Escherichia coli was placed in a constant temperature biological culture cabinet and kept in an inverted state for 24 h. Colonies with good colony growth were selected and transplanted into the medium. The culture medium without transplanted colonies was irradiated and sterilized with a UV lamp for 12 h, and then sealed in a sub–packing bag as a blank control group.

In view of the difference in the properties of dressing-applied fiber materials, the bacterial colonies in the medium were diluted to 10^−6^, and, according to the actual reaction time, the growth in the number of colonies of staphylococcus units in the unit time was controlled to be 30 cfu. After the culture treatment was completed, the bacterial solution was evenly applied to the medium utensils with the coating rod. The colonies in the culture medium were used as the treatment object of the polysaccharide-based degradable hydrogel wound dressing, and the inhibitory rate of the wound dressing was measured according to the number of inhibited colonies within the cycle range. According to the colony medium prepared above, the bacteriostatic rate of hydrogel wound dressing was defined, and the numerical relationship is expressed as [25]:(2)$$Y=\frac{W_{b}-W_{c}}{W_{b}}\times 100\%$$
where Y is the calculated bacteriostasis rate; W_b_ is the number of colonies implanted in the medium; and W_c_ is the number of colonies in the control group. According to the numerical relationship defined above, the bacteriostatic effect was tested in an Escherichia coli environment and a staphylobacter colony at 4 h intervals; the bacterial inhibition results of wound dressings were finally obtained, as shown in Figure 6.

As can be seen from the bacterial inhibition results of wound dressings in Figure 6a,b, the polysaccharide–based degradable hydrogel wound dressing has a better bacteriostatic effect in the environments of E. coli bacteria and staphylobacter. Figure 6c shows wound dressings can effectively inhibit 53% of bacteria, and the bacterial inhibition effect of wound dressings is better. In the bacterial environment of Staphylococcus, the wound dressing can effectively inhibit 47% of the bacteria, and the wound dressing has a better bacterial inhibition effect. In conclusion, the wound dressing produced has a better bacterial inhibition effect, and can be used in the process of wound care.

### 3.4. Tissue Collagen Deposition

MTT analysis was used to evaluate the application of the prepared wound dressings according to the bacteriological inhibition effect described above. A small number of EA.hy926 endothelial cells were taken and placed on the membrane dressing and cultured for 48 h. Under sterile conditions, a wound was manually marked on the mold setting, and the collagen distribution of the tissue on the wound was observed with an electron microscope, as shown in Figure 7.

As can be seen from the collagen distribution of the wound in Figure 7a, no bacterial toxicity was present on the surface of the wound, and no collagen proliferation occurred on the wound. After the prepared hydrogel wound dressing was placed on the wound, it was observed for 12 h. According to Figure 7b, part of the tissue collagen deposition was found in the epidermal structure. With the continuous enhancement of the bacterial inhibition effect of the hydrogel wound dressing, the wound epidermis can absorb protein, produce fibronectin in the tissue structure, promote the deposition of the collagen structure, and accelerate hemostasis and wound healing. After further collagen activity of the tissues, collagen deposition after 24 h of application of the wound dressing was observed, and the specific results are shown in Figure 7c. As shown in Figure 7c, as the treatment time of wound dressing increases, a phenolic substance is generated between the activated tissue collagen and the dressing, which promotes the regeneration of tissue collagen, activates the cells in the wound, and achieves the rapid deposition of the collagen structure. Therefore, the prepared polysaccharide–based degradable hydrogel wound dressing can effectively inhibit bacterial regeneration, promote collagen deposition on the wound surface, and accelerate wound healing. This has excellent clinical application in the course of medical nursing.

## 4. Conclusions

In this study, a polysaccharide–based biodegradable hydrogel was synthesized and applied to treat dermal wounds. In vivo results showed that the hydrogel can accelerate granulation tissue formation, increase collagen deposition, and thus promote the wound closure. The results show that the polysaccharide-based biodegradable hydrogel has degradability and excellent antibacterial properties. Furthermore, by simulating the wound environment, it was found that the synthetic hydrogel wound dressing can effectively inhibit bacteria and accelerate the healing of the wound surface.

## Figures and Tables

**Figure 1 materials-16-01358-f001:**
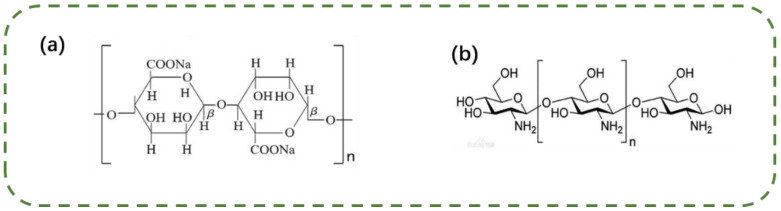
Chemical structures of (**a**) SA and (**b**) CS.

**Figure 2 materials-16-01358-f002:**
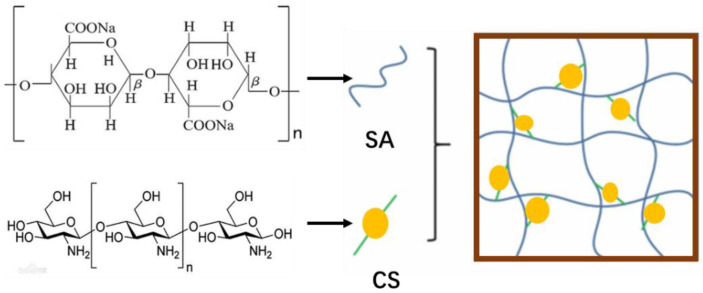
Schematic diagram of polysaccharide–based hydrogel synthesis.

**Figure 3 materials-16-01358-f003:**
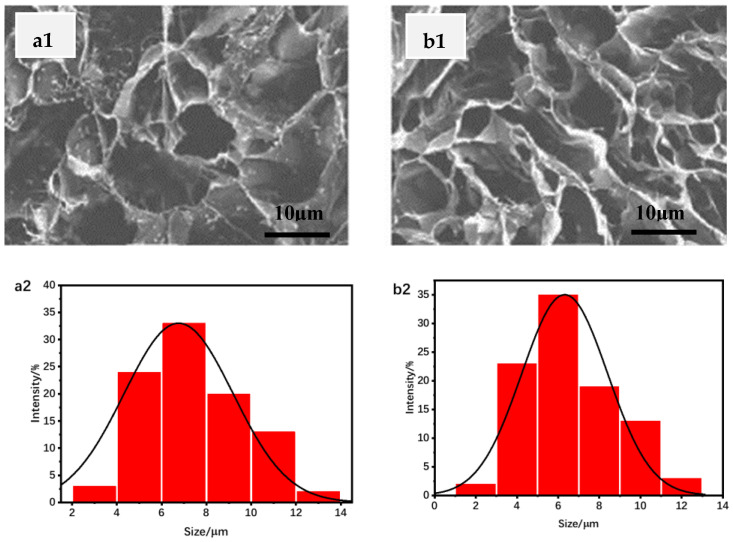
(**a1**,**b1**) SEM images of hydrogel structure, (**a2**,**b2**) particle size distributions of the hydrogel structure.

**Figure 4 materials-16-01358-f004:**
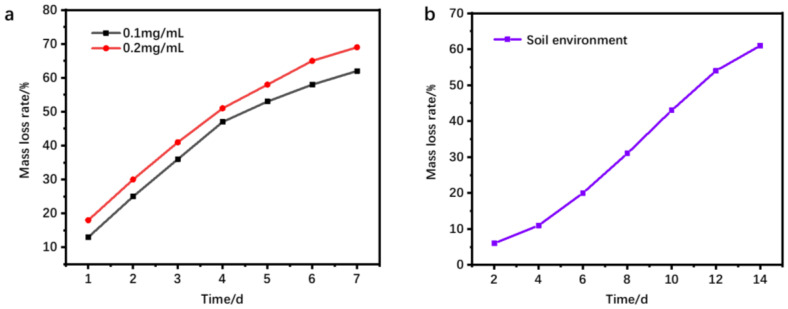
The mass loss of hydrogels with trypsin (**a**) and the mass loss of hydrogels in the soil environment (**b**).

**Figure 5 materials-16-01358-f005:**
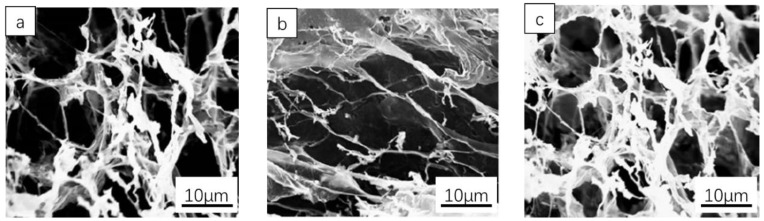
SEM of the hydrogel with trypsin at concentrations of 0.1 mg/mL (**a**) and 0.2 mg/mL (**b**) after 4 days, and of the hydrogel in the soil environment (**c**) after 7 days.

**Figure 6 materials-16-01358-f006:**
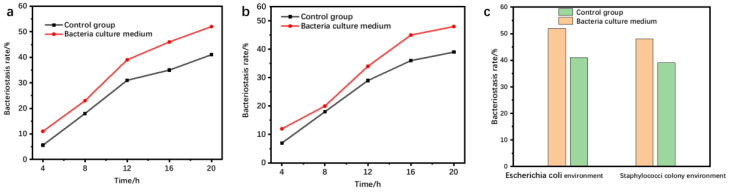
Bacterial inhibition results of wound dressing. (**a**) The bacteriostatic effect of Escherichia coli varying with time, (**b**) the bacteriostatic effect of staphylobacter varying with time, (**c**) comparison of final bacteriostatic effect.

**Figure 7 materials-16-01358-f007:**
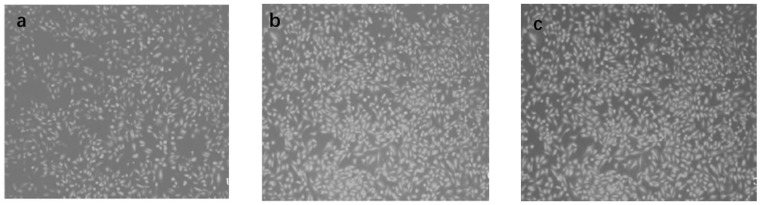
(**a**) Collagen distribution in the wound, (**b**) collagen deposition after 12 h of dressing treatment, (**c**) collagen deposition in tissues after 24 h of dressing treatment.

**Table 1 materials-16-01358-t001:** Swelling rate of wound dressings.

Temperature (°C)	30	35	40	45	50	60	70
Swelling rate (%)	26.8	31.2	34.2	38.9	42.5	47.6	53.7

## Data Availability

Not applicable.

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
