# Peer review of "Synthesis and Properties of Biodegradable Hydrogel Based on Polysaccharide Wound Dressing"

_materials, 2023, doi:10.3390/ma16041358_

Round 1
Reviewer 1 Report
COMMENTS TO THE AUTHOR(S)
Meiling Shao and group propose research on the “Synthesis and properties of biodegradable hydrogel based on polysaccharide wound dressing”. But, this manuscript is not suitable for publication, as detailed below:
1. The entire article has a lot of grammatical errors, and the scientific language of the overall article is also not up to the mark.
2. Abstract is very generalized. So, it is suggested that authors must rewrite the informative abstract.
3. Introduction section of this research is also very poor. Authors must rewrite and provide detailed information about the utilized polysaccharides. The overall manuscript is very generalized and has poor results.
4. Author dissolves the chitosan in the water. Therefore, the materials section must mention the proper information about the polysaccharides like molecular weight and other specifications.
5. Authors used the old literature for reference. Therefore, the authors should use more recent research to reference this study.
Author Response
We feel great thanks for your professional review work on our article. As you are concerned, there are several problems that need to be addressed. According to your nice suggestion, we have made extensive corrections to our previous draft, detailed corrections are listed below.
1.The article has been polished to correct grammar and language problems.
2.Abstract has been revised.
3.Enriched the content of the introduction.
4.The information about the polysaccharides have added to paper.
5.We have used more recent research to reference this study.

Reviewer 2 Report
The paper "Synthesis and properties of biodegradable hydrogel based on polysaccharide wound dressing" is interesting. It can be accepted for publication if it corrects the points below:
At Abstract-Repharase this passage:
"Herein, a biodegradable hydrogel based on polysaccharide was synthesized and applied as wound dressing. And characterize the swelling degree and degradability of hydrogel as the characteristics of wound dressing. "
At Introduction -Check how to write the bibliographic reference in all the text.
ex:
"A variety of hydrophilic groups are attached to the polymer side chain skeleton in the hydrogel, including –COOH,-OH,-SO3H,-NH2,-CONH and so on, which can maintain the water retention property of hydrogel. 3.As a kind of extracellular matrix(ECM) polymer, hydrogel has good biocompatibility, and has been widely used in biomedicine, tissue engineering and other fields4."
At 2. Materials and methods: Corect word:
"The cell cultrue plates were purchased from Fuzhou Aoyan Experimental Equipment Co., Ltd, Fuzhou, China."
At 2.2 Methods
a) Edit the text
"After centrifugation, the precipitation was dispersed in the liquid, and sodium hydroxide with a mass concentration of 150 mg/mL was added to adjust the solvent to be alkaline"
Pay attention to noting the figures (ex. fig. 1)
At 3.1 Swelling degree analysis:
Put the units of measure in parentheses( ex tab 1);
The temperatures in the text do not correspond to those in table 1
It refers to tab 3, but there is no such thing:
“It can be seen from Table 3 that the prepared biodegradable polysaccharide…”
At 3.2 Hydrogel degradability,
Uniform text rewriting of Fig or Figure( ex Fig2a , Figure2a);
The text talks about hydrogel with polysaccharide and the figure shows the SEM of a hydrogel with trypsin ( fig 3)
"Fig3. SEM of the hydrogel with trypsin at a concentration of 0.1mg/mL (a) and 0.2mg/mL (b) after 4 days, the hydrogel in the soil environment (c) after 7 days.";
At 3.3 Bacteriostatic performance, check the content of the text ” Medium selected nutrient AGAR medium, prepare 5 g beef paste, add 10 g peptone, stir evenly, place 5 g sodium chloride, add appropriate amount of distilled water, dissolve into 1 000 mL solution. Then add a concentration of 0.1 mol/L sodium peroxide, adjust the P-H value of the solution to weak acid, put in the sterilization box sterilization and then treatment for 30 min.”
“A small number of EA. hy926 endothelial cells…” Check the correct spelling of “EA.hy926 ”
At 3.4 Tissue collagen deposition, check the content of the text:
“After further collagen activity of the tissues, collagen deposition after 24 h application of the wound dressing was observed, and the specific results were shown in Figure5c. As shown in Figure3c, as the treatment time of wound dressing increases, a phenolic substance is generated between the activated tissue collagen and the dressing, which promotes the regeneration of tissue collagen, activates the cells in the wound, and achieves the rapid deposition of collagen structure.”
Author Response
We feel great thanks for your professional review work on our article. As you are concerned, there are several problems that need to be addressed. According to your nice suggestion, we have made extensive corrections to our previous draft, detailed corrections are listed below.
Corrections have been made in yellow in the text.

Reviewer 3 Report
This manuscript can be accepted in this form.
Author Response
We feel great thanks for your professional review work on our article. As you are concerned, there are several problems that need to be addressed. According to your nice suggestion, we have made extensive corrections to our previous draft.
Reviewer 4 Report
This work entitled “Synthesis and properties of biodegradable hydrogel based on polysaccharide wound dressing” is an article reporting the design of a biodegradable polysaccharide based hydrogel which was applied as wound dressing. The authors investigated the mechanism of the hydrogel on wound healing and it was found that the hydrogel could effectively inhibit bacteria and accelerate wound healing.
The manuscript needs to be improved:
- Abstract: rewrite better the phrase “And characterize the swelling degree and degradability of hydrogel as the characteristics of wound dressing.”
- 2.2. Methods: the authors write: “SA and CS with the ratio of 2/1 was dissolved in autoclaved deionized water to obtain a final concentration of 40 mg/ml.”. Define the units in the 2/1 ratio (mass or volume?).
- Page 3: the authors write somewhere “The mixture solution was introduced into the mold and polymerized to obtain…”. What is the mixture solution? SA, CS, HA, sodium periodate and ethylene glycol? The experimental procedure at this point should be a bit clearer. Surely, a scheme representing the synthesis of the polysaccharide based degradable hydrogel with structures of the reagents and the mechanisms how the reagents react to form the hydrogel is important and necessary.
- The authors need to check how they call Tables and Figures in the whole text. Let me explain what I mean: for example, we can see somewhere written “Figure 3” and somewhere else “Fig.3”. Similarly, “Tab.1” and “Table 1”. Keep a uniform format.
- 3.3 Bacteriostatic performance: Maybe the first paragraph fits better in the ‘2. Materials and methods’ section.
- Page 7: rephrase the sentence “As can be seen from the bacterial inhibition results of wound dressings in Figure 4(a) and (b) polysaccharide based degradable hydrogel wound dressing has a faster bacteriostatic effect in the environment of E. coli bacteria and staphylobacter varies”, as in the end of it is confusing.
- Figures 5b, 5c: It seems that these two pictures are the same, what is the difference between 12 and 24 h of collagen deposition? What is the conclusion?
Finally, the authors should check again the whole text as the English language needs to be polished.
Author Response
We feel great thanks for your professional review work on our article. As you are concerned, there are several problems that need to be addressed. According to your nice suggestion, we have made extensive corrections to our previous draft, detailed corrections are listed below.
Corrections in the text have been highlighted in yellow.
Figures 5b, 5c: Figure5c is darker than 5b, indicating the production of phenols.

Round 2
Reviewer 1 Report
Accepted in present form
Author Response
We feel great thanks for your professional review work on our article. As you are concerned, there are several problems that need to be addressed. According to your nice suggestion, we have modified it accordingly.
Reviewer 4 Report
Thank the authors for revising their manuscript. There are still though, things that have not been revised and according to me need to be taken care of so the text will be more understandable.
- Abstract: rephrase “And characterize the swelling degree and degradability of hydrogel as the characteristics of wound dressing.” It is not correctly written. Maybe “And the swelling degree and degradability of hydrogel were characterized as the characteristics of wound dressing.” is a better choice….
- Figure 1: the chemical structure of CS is known, or someone can easily find it on the internet.
The manuscript would be better if a Scheme or Figure showed the mechanism how the reagents chemically react each other to form the hydrogel. What is the chemistry behind their product (what bonds etc.).
- Page 7: in the sentence “As can be seen from the bacterial inhibition results of wound dressings in Figure 4(a) and (b) polysaccharide based degradable hydrogel wound dressing has a faster bacteriostatic effect in the environment of E. coli bacteria and staphylobacter varies”, delete the word varies which has been wrongly copied-pasted from the Figure caption Fig.5b.
Author Response
We feel great thanks for your professional review work on our article. As you are concerned, there are several problems that need to be addressed. According to your nice suggestion, we have made extensive corrections to our previous draft, detailed corrections are listed below.
1.The abstract has been revised as you suggested.
2. A hydrogel has been added to generate a schematic.
3. The "varies" has been wrongly copied-pasted from the Figure caption Fig.5b.